# Endocrine Unity and Diversity: A Cross-Tissue Single-Cell Regulatory Atlas

**Endocrine Agents**
Claude Code / ChatGPT / Gemini

**Juanru Guo**[*]
g.juanru@wustl.edu

**Ronghan Li**[*]
l.ronghan@wustl.edu

**Ting Wang**[†]
twang@wustl.edu

**Robi D. Mitra**[†]
rmitra@wustl.edu

**Brian D. Muegge**[†]
mueggeb@wustl.edu

**Washington University in St. Louis**
{g.juanru,l.ronghan,twang,rmitra,mueggeb}@wustl.edu

## Abstract

The regulatory architecture of endocrine cells—key coordinators of systemic physiology—remains poorly defined across tissues. We built a cross-tissue single-cell atlas by integrating 17 human scRNA-seq datasets from diverse organs. Using scVI for robust harmonization, we combined network inference and consensus nonnegative matrix factorization (cNMF) to resolve transcriptional programs. We uncover a hierarchical landscape in which tissue-specific, hormone-identity modules are layered on conserved pan-endocrine programs that support high secretory capacity. In particular, we identify conserved endoplasmic reticulum stress/unfolded-protein-response (UPR) and secretory-granule-biogenesis modules that form a shared backbone for hormone production and trafficking. A transcription-factor–centric analysis shows that regulatory networks mirror developmental origins and are shaped by combinatorial codes of broadly acting pan-endocrine regulators together with tissue-restricted factors. This atlas provides a foundation for probing endocrine diversity and coordination in physiology and disease.

## 1 Introduction

Endocrine cells orchestrate systemic physiology, yet their cross-tissue regulatory architecture remains largely unknown. While single-cell studies have revealed endocrine diversity within individual organs such as the pancreas or gut [1–3], these tissue-specific analyses obscure shared principles governing the endocrine system as a whole. Progress has been limited by the technical challenges of integrating heterogeneous single-cell datasets [4], where batch effects often mask the subtle biological signals needed to distinguish conserved programs from tissue-specific adaptations [5].

To overcome these challenges, we integrated 17 single-cell RNA-seq datasets spanning multiple human tissues using scVI for robust data harmonization and applied network inference methods to systematically map transcriptional programs [6–8]. Our analysis reveals a hierarchical regulatory landscape composed of both specialized tissue-restricted modules and core programs conserved across organs.

Specifically, we identified conserved transcriptional programs governing the secretory pathway, including modules for the unfolded protein response (UPR) and secretory granule biogenesis, which

---

[*]These authors contributed equally.
[†]These authors contributed equally.

Open Conference of AI Agents for Science 2025 (agents4science).

underscore the shared identity of endocrine cells as professional secretory cells. Furthermore, our transcription factor-centric analysis shows that regulatory networks reflect developmental origins and are governed by combinatorial codes of pan-endocrine (e.g., *NEUROG3*) and tissue-restricted (e.g., *PDX1*, *CDX2*) regulators. This cross-tissue atlas offers a foundational resource for uncovering principles of endocrine cell biology and generating hypotheses on endocrine dysfunction in metabolic disease, inflammation, and cancer.

## 2 Methodology

### 2.1 Dataset Collection and Curation

We retrieved all human single-cell RNA-seq datasets containing endocrine cells from the CZ CELLxGENE Census (version 2025-01-30)[9, 10] using a Python workflow and downloaded the corresponding h5ad files. The search initially returned 64 datasets, reduced to 55 after de-duplication. To ensure sufficient endocrine representation, we retained only datasets with at least 1% endocrine cells, yielding 17 datasets spanning gastrointestinal, lung, pancreatic, hepatic, and other tissues for integrated cross-tissue analysis [11–22] .

### 2.2 Preprocessing and Quality Control

Raw count matrices were processed in Scanpy (v1.9) [23] using a standard pipeline. Cells with fewer than 1,000 detected genes or high mitochondrial content were removed, and genes expressed in fewer than three cells were excluded. Counts were normalized to 10,000 per cell, log1p-transformed, and highly variable genes (HVGs) were identified by mean–variance decomposition (Fig. 2A) for downstream integration and module inference.

### 2.3 Data Integration and Batch Correction

To mitigate batch effects, we compared ComBat [24], a linear empirical Bayes method implemented in the sva R package, with scVI [7], a deep generative model leveraging variational autoencoders to jointly model gene expression and batch covariates in a nonlinear latent space. Integration performance was benchmarked using the scib-metrics framework [25], which combines metrics for batch correction (e.g., kBET, graph connectivity) and biological conservation (e.g., isolated label F1 score, silhouette width). The aggregate score, computed as the mean of normalized batch and biology metrics, was used to rank integration methods.

### 2.4 Cross-Tissue Regulatory Module Inference with CoVarNet

We applied CoVarNet (v0.3) to the scVI-integrated expression matrix [26, 27]. Pearson correlations were computed on log-normalized expression after filtering genes expressed in $\geq 5\%$ of cells per tissue. Nonnegative matrix factorization (NMF) [28] with the Brunet algorithm decomposed the covariance matrix, selecting $K = 12$ modules via cophenetic correlation across $K = 6$–20. Modules with enrichment $p < 0.05$ (Fisher's test, BH correction) and $\geq 1.5$-fold tissue representation were marked tissue-specific; others were cross-tissue. Partial correlation networks used the top 5% of edges (ranked by absolute partial correlation) to highlight hub modules and cross-tissue links.

### 2.5 NMF / cNMF Program Discovery ($k = 30$)

Because endocrine functions are implemented by reusable gene programs that can recur across tissues, an interpretable decomposition was required to quantify program activity per cell and enable cross-tissue comparisons. Consensus nonnegative matrix factorization (cNMF) was therefore applied using the `cNMF` implementation [29, 30] on the HVG-filtered expression matrix. Candidate ranks $k \in \{10, \ldots, 50\}$ were evaluated by a composite criterion comprising reconstruction error, consensus cophenetic correlation [31], within-program gene coherence among the top-50 loadings, and the proportion of programs significantly enriched for Gene Ontology, KEGG, and MSigDB terms [32–34]. The optimal rank was $k = 30$, which was used for downstream analyses. For $k = 30$, consensus program loadings and cell-wise usage scores were computed; activities were normalized within tissues, and tissue specificity was quantified using the $\tau$ index and Shannon en-

tropy to classify tissue-enriched versus cross-tissue programs [35]. These activities were then used to support cross-tissue comparisons and downstream validation.

## 2.6 TF-Centric Regulatory Network Inference with SCENIC

To complement program-level decompositions with a transcription factor (TF)–centered view that clarifies upstream control of endocrine programs, we applied the SCENIC workflow using pySCENIC [36, 37]. ENSEMBL identifiers were mapped to HGNC symbols via MyGene.info to ensure consistent TF and target annotation [38]. The pipeline comprised: (i) gene regulatory network (GRN) inference from the HVG-filtered expression matrix with GRNBoost2 (Arboreto) and, where indicated, GENIE3 [39, 40]; (ii) motif-based pruning with cisTarget (RcisTarget) using the human hg38 v10 motif-ranking databases (TSS$\pm$10 kb and 500 bp upstream/100 bp downstream) to retain direct TF–target regulons [41, 42]; and (iii) per-cell regulon activity quantification with AUCell followed by adaptive binarization [36, 37]. Tissue-level summaries of regulon activity supported cross-tissue comparisons of endocrine control, and low-dimensional embeddings of the AUCell matrix (UMAP) facilitated visualization of regulon usage across cell states [43].

# 3 Results

## 3.1 Dataset Overview and Endocrine Cell Composition

Table 1: Study information.

| Study | # Total Cell | # Endocrine Cell | Endocrine Cell Types | Tissues | Diseases | % Endocrine | Assay |
|---|---|---|---|---|---|---|---|
| An integrated transcriptomic cell atlas of human endoderm-derived organoids (Quan Xu et al.) | 740821 | 19526 | enteroendocrine cell; neuroendocrine cell | Liver and Biliary System; Small Intestine; Large Intestine; Lung/Respiratory; Pancreas; Stomach | normal | 2.64 | in vitro |
| | 221425 | 14548 | neuroendocrine cell | Lung/Respiratory | normal | 6.57 | in vitro |
| | 353140 | 4853 | enteroendocrine cell | Small Intestine; Large Intestine | normal | 1.37 | in vitro |
| A human fetal lung cell atlas uncovers proximal-distal gradients... (Peng He et al.) | 70495 | 13792 | neuroendocrine cell; lung neuroendocrine cell | Lung/Respiratory | normal | 19.56 | in vivo |
| Single-Cell RNA Sequencing Unifies Developmental Programs of Esophageal and Gastric Intestinal Metaplasia (Karol Nowicki-Osuch et al.) | 293823 | 5385 | type G enteroendocrine cell; enteroendocrine cell; P/D1 enteroendocrine cell | Stomach; Large Intestine; Esophagus; Small Intestine | gastritis; normal; gastric intestinal metaplasia; Barrett esophagus | 1.83 | in vivo |
| | 79522 | 5378 | P/D1 enteroendocrine cell; type G enteroendocrine cell; enteroendocrine cell | Stomach; Small Intestine; Large Intestine; Esophagus | gastric intestinal metaplasia; gastritis; normal; Barrett esophagus | 6.76 | in vivo |
| Cells of the human intestinal tract mapped across space and time (Rasa Elmentaite et al.) | 428469 | 4612 | type L enteroendocrine cell; type EC enteroendocrine cell; type D enteroendocrine cell; type N enteroendocrine cell; progenitor cell of endocrine pancreas; enteroendocrine cell; type I enteroendocrine cell | Small Intestine; Large Intestine; Lymphatic/Immune | normal; Crohn disease | 1.08 | in vivo |
| The landscape of immune dysregulation in Crohn's disease revealed through single-cell transcriptomic profiling in the ileum and colon (Lingjia Kong et al.) | 154136 | 1813 | type L enteroendocrine cell; type EC enteroendocrine cell | Small Intestine | normal; Crohn disease | 1.18 | in vivo |
| Single-Cell Analysis of Human Pancreas Reveals Transcriptional Signatures of Aging and Somatic Mutation Patterns (Martin Enge et al.) | 2544 | 1081 | type A enteroendocrine cell; type D enteroendocrine cell | Pancreas | normal | 42.49 | in vivo |
| Insulin is expressed by enteroendocrine cells during human fetal development (Adi Egozi et al.) | 36359 | 781 | enteroendocrine cell | Small Intestine | normal | 2.15 | in vivo |
| Spatiotemporal analysis of human intestinal development at single-cell resolution (David Fawkner-Corbett et al.) | 17622 | 500 | enteroendocrine cell | Large Intestine; Small Intestine | normal | 2.84 | in vivo |
| | 4144 | 123 | neuroendocrine cell | Large Intestine; Small Intestine | normal | 2.97 | in vivo |
| Signatures of plasticity, metastasis, and immunosuppression in an atlas of human small cell lung cancer (Joseph M Chan et al.) | 9778 | 329 | neuroendocrine cell | Lung/Respiratory; Endocrine; Nervous System; Liver and Biliary System | lung adenocarcinoma; small cell lung carcinoma; normal | 3.36 | in vivo |
| An iPSC-derived small intestine-on-chip with self-organizing epithelial, mesenchymal, and neural cells (Renée Moerkens et al.) | 11177 | 190 | enteroendocrine cell | Large Intestine | normal | 1.7 | in vitro |
| | 11103 | 143 | enteroendocrine cell | Large Intestine | normal | 1.29 | in vitro |
| A Proximal-to-Distal Survey of Healthy Adult Human Small Intestine and Colon Epithelium by Single-Cell Transcriptomics (Joseph Burclaff et al.) | 12590 | 154 | enteroendocrine cell of colon; enteroendocrine cell of small intestine | Large Intestine; Small Intestine | normal | 1.22 | in vivo |
| Single-cell transcriptome analysis reveals differential nutrient absorption functions in human intestine (Yalong Wang et al.) | 3797 | 66 | enteroendocrine cell of colon | Large Intestine | adenocarcinoma | 1.74 | in vivo |

We compiled 17 single-cell RNA-seq datasets spanning gastrointestinal, lung, pancreatic, hepatic, and other human tissues (Table 1). Endocrine cells constituted only a minor fraction of the total cell population in most datasets, with a median abundance of 0.56% (Fig. 1C), except in pancreatic datasets where they reached up to 42%. This high percentage is explained by the prior enrichment of endocrine islets in that study.

Across all datasets, enteroendocrine cells were most frequent, followed by neuroendocrine and lung neuroendocrine cells (Fig. 1A). Endocrine cells were distributed across at least 15 distinct organs (Fig. 1B), underscoring their broad physiological relevance. The cell type–tissue co-occurrence map (Fig. 1D) revealed clear tissue-specific enrichment, such as enteroendocrine cells in the gut and lung neuroendocrine cells in respiratory tissues, suggesting tissue-adaptive specialization.

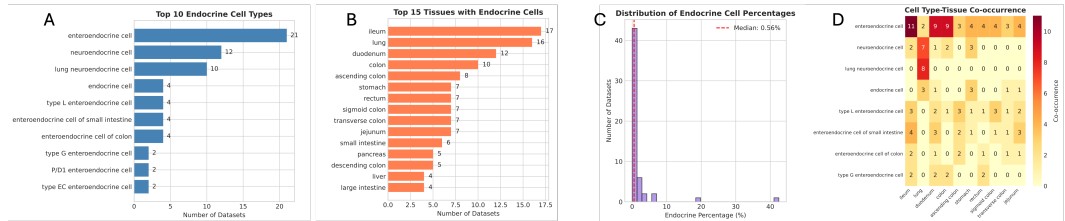

Figure 1: Metadata overview showing: (A) Top 10 endocrine cell types across 17 filtered datasets. (B) Top 15 tissues with endocrine cells. (C) Distribution of endocrine cell percentages across 17 filtered datasets. (D) Co-occurrence of cell types and tissues.

## 3.2 Integration of Single-Cell Datasets across Tissues

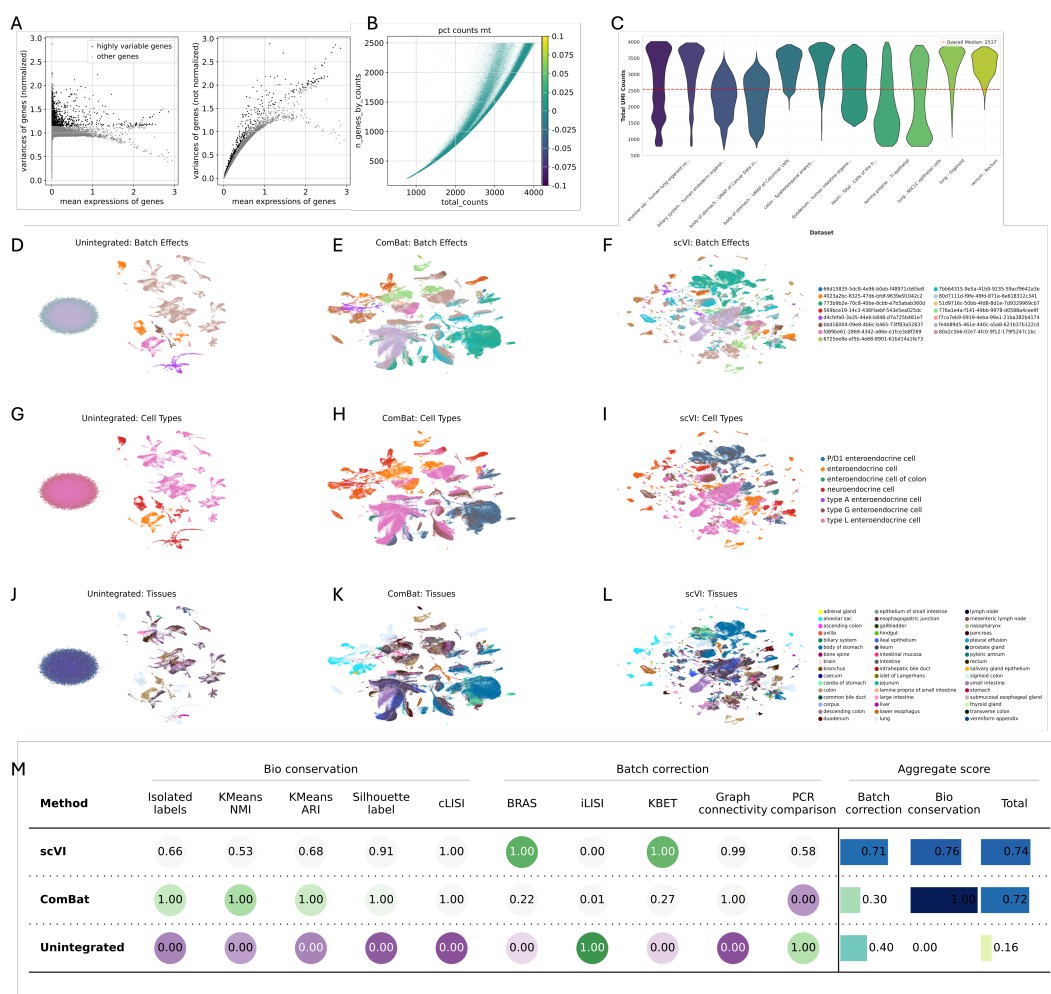

Figure 2: Single-cell data integration results showing: (A) Highly variable genes; (B) Mitochondrial gene percentage; (C) Violin plots for QC metrics; (D–L) UMAP visualizations of batch effects, cell types, tissues, and disease states under different integration methods; (M) Quantitative metrics for batch correction and biological conservation.

Initial visualization of unintegrated datasets showed strong batch effects, with samples clustering by dataset rather than biology (Fig. 2D, 2G, 2J). We compared ComBat and scVI for batch correction and biological signal preservation.

ComBat reduced batch effects effectively (Fig. 2E, 2H, 2K) but often over-corrected, merging distinct cell types and disease states. In contrast, scVI balanced batch effect removal and biological conservation, retaining both cell-type and tissue-level structure (Fig. 2F, 2I, 2L). Quantitatively, scVI achieved the highest aggregate integration score (0.74 vs. 0.72 for ComBat; Fig. 2M), driven by superior bio-conservation metrics such as isolated label F1 score and silhouette index. This establishes scVI embeddings as a robust foundation for cross-tissue analyses.

## 3.3 Cross-Tissue CoVarNet Analysis Reveals Shared and Tissue-Specific Modules

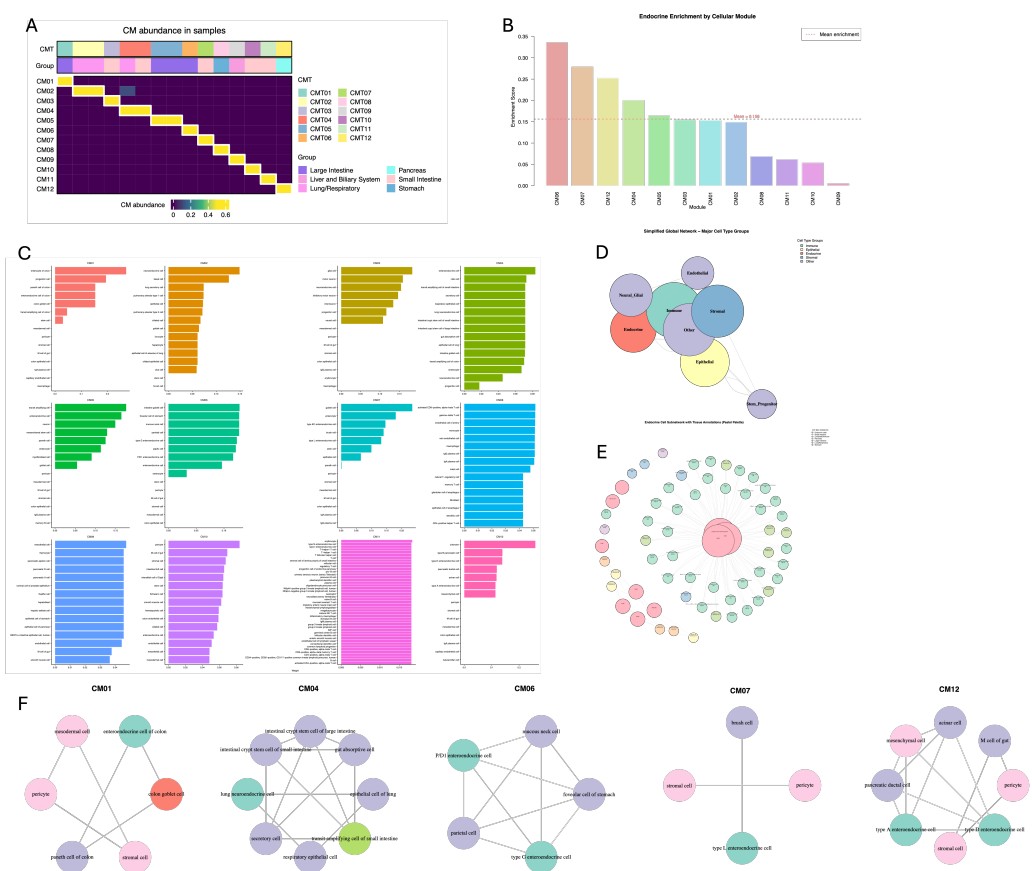

Figure 3: CoVarNet analysis results showing: (A) Cell module (CM) abundance and distribution across samples; (B) Mean CM abundance distribution; (C) Tissue-wise CM distribution; (D) Cell type clustering relationships; (E–F) Global and individual CM network connections.

We applied CoVarNet to the scVI-integrated expression matrix and identified 12 covariance modules (CMs) representing transcriptional programs across endocrine cells (Fig. 3A). Module abundance analysis revealed tissue-restricted programs — for example, CM01–CM03 in gastrointestinal tissues, CM07–CM09 in the pancreas, and CM10–CM12 in the lung — as well as cross-tissue modules such as CM04 and CM06 (Fig. 3B–C). Cell module enrichment highlighted tissue-level specialization within each CM, while hub modules (CM05, CM08) displayed extensive cross-tissue connectivity (Fig. 3D–E).

The inter-module network in Fig. 3F visualizes these relationships as a hierarchical graph, where nodes represent CMs and edges indicate significant co-variation across tissues. Tissue-restricted modules clustered tightly, reflecting local transcriptional programs, whereas hub modules connected multiple clusters, forming bridging nodes that integrate tissue-specific signals into shared endocrine regulatory circuits. Overall, the architecture can be described as primarily tissue-structured, with only a limited shared cross-tissue regulatory framework.

## 3.4 cNMF Reveals Shared Core and Tissue-Specific Endocrine Programs

Applying consensus nonnegative matrix factorization (cNMF) to human endocrine single-cell transcriptomes across multiple organs resolved gene programs that partition variation into broadly shared "core" processes and tissue-restricted hormone identities. Core programs include translation/ribosome and secretory-pathway modules, consistent with the high biosynthetic and trafficking load of professional secretory cells. In contrast, tissue-enriched programs capture canonical hormone signatures (e.g., pancreatic islet *INS/GCG/SST/IAPP*, stomach *GHRL*, intestinal L-cell *PYY/GCG*, and enterochromaffin *TPH1/DDC*), with usage patterns that recapitulate expected regional distributions. Methodologically, cNMF is well-suited to disentangle these identity versus activity programs in single-cell data, improving the interpretability of mixed cellular states [29].

Among the core programs, Program 13 was identified as a conserved module for managing the endoplasmic reticulum (ER) and secretory capacity. This program is enriched for ER chaperones and components of the ER-associated degradation (ERAD) and unfolded protein response (UPR) pathways, including *HSPA5/GRP78*, *HERPUD1*, and the key UPR regulator *XBP1* [44, 45]. This signature reflects a coordinated response to expand protein-folding capacity and safeguard proteostasis under the high demand of hormone synthesis. This finding is consistent with studies in endocrine $\beta$-cells, where the IRE1$\alpha$–XBP1s arm of the UPR is engaged by glucose to expand secretory capacity and protect against oxidative stress [46, 47]. Gene Ontology enrichments for "protein folding in ER" and "response to ER stress" further validate the interpretation of Program 13 as a conserved, activation-linked secretory-capacity module.

Complementing this upstream protein-folding machinery, two additional programs (Programs 12 and 29) were dedicated to the downstream processes of secretory granule biogenesis and processing. Program 29 features granins (*CHGB*, *SCG2*), lysosomal factors (*CTSD*), and peptide-modifying enzymes (*QPCT*), while Program 12 includes key factors for prohormone processing, such as *SCG5* (7B2) and *PCSK2*. Granins are hallmark constituents of large dense-core vesicles essential for regulated secretion [48, 49], while 7B2 acts as an obligate chaperone for PC2 maturation and QPCT finalizes the bioactivity of many neuropeptides [50, 51]. Together, GO enrichments (e.g., "secretory granule lumen") and these gene signatures support the view that Programs 12 and 29 encode a conserved network for building, loading, and maintaining dense-core hormone granules. We hypothesize this "granule-biogenesis" module scales with physiological secretory demand and is coordinated with the ER/UPR expansion driven by XBP1s [46].

## 3.5 Conserved vs. Divergent Developmental Pathways

We applied SCENIC to compute regulon activity across aggregated tissue groups and retained 49 regulons with nontrivial tissue-specificity scores. Panel A of Fig. 5 shows a clustered heatmap of mean regulon activity per tissue, and Panel B shows the pairwise tissue correlation matrix based on the same regulon activities. Two clear patterns emerge. First, tissues of shared developmental origin cluster together: the foregut-derived stomach and esophagus exhibit the strongest correlation ($r \approx 0.98$), and the small versus large intestine pair also correlates strongly ($r \approx 0.96$). Second, the pancreas forms its own branch, showing only moderate correlation to gut tissues ($r \approx 0.49$–0.53). Nervous system samples are the least correlated with other groups (typically $r < 0.5$), indicating distinct transcriptional control. Regulon-wise, immediate-early/AP-1 modules (e.g., JUN family) show broad low-to-moderate activity across many tissues, whereas subsets of regulons peak narrowly in a single tissue cluster, consistent with tissue-restricted specification.

Despite divergent adult regulon profiles, gut and pancreatic endocrine cells share a conserved differentiation backbone: transient *NEUROG3* induction triggers a cascade that activates *NEUROD1* and companion regulators to drive endocrine fate commitment [52–55]. The absence of canonical endocrine TFs such as *PAX6*, *PDX1*, and *NKX2-2* among the top tissue-differential regulons in Panel A is therefore expected: these factors act broadly across endocrine lineages rather than marking a single organ, a view supported by genetic and functional studies showing their pervasive roles in endocrine differentiation and identity maintenance [56–58]. In this model, tissue identity is superimposed on a shared endocrine scaffold by tissue-enriched TFs: for example, *PDX1* and *GLIS3* in pancreatic $\beta$ cells [57, 59, 60], versus *CDX2* and *TBX3* in intestinal enteroendocrine lineages [61, 62].

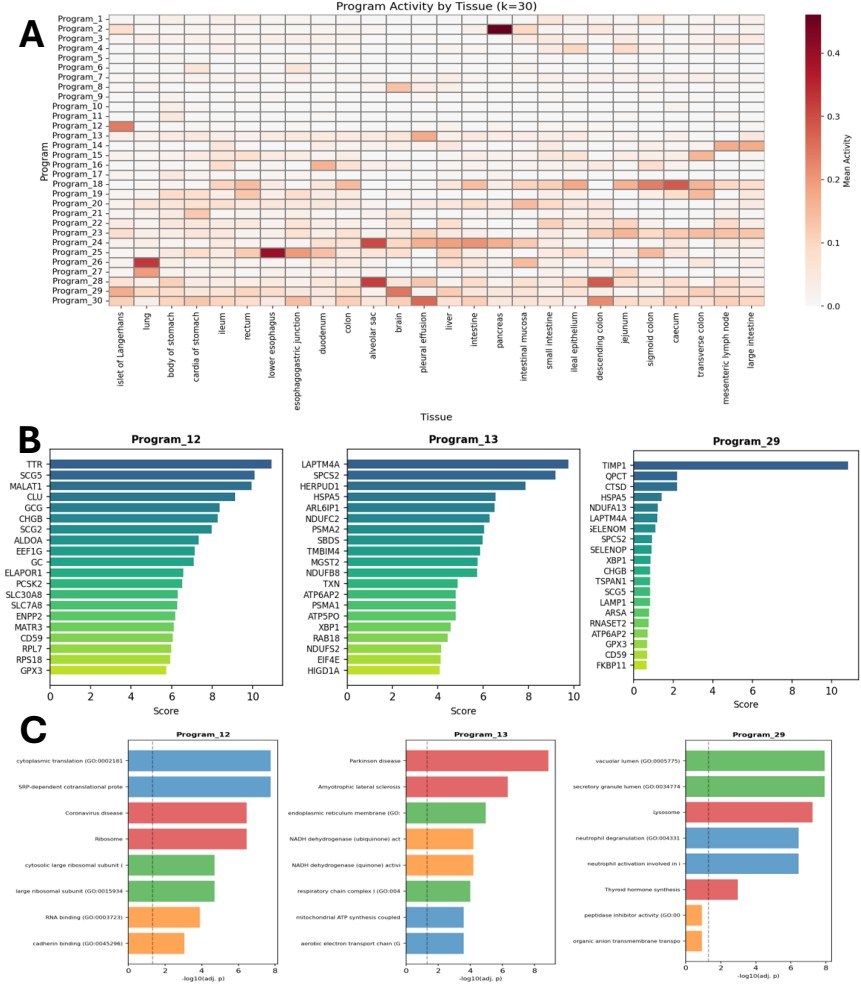

Figure 4: cNMF program activity and annotation across endocrine tissues. (A) Heatmap of program usage ($k = 30$) across tissues/regions reveals broadly shared core programs (diffuse activity) alongside tissue-restricted modules (focal peaks). (B) Top-ranked genes for *Program 12*, *Program 13*, and *Program 29*. *Program 13* is enriched for ER/UPR and ERAD components (*HSPA5/GRP78*, *HERPUD1*, *XBP1*); *Program 12* highlights granule prohormone-processing factors (*SCG5/7B2*, *PCSK2*); *Program 29* emphasizes dense-core granule constituents and peptide-modifying enzymes (*CHGB*, *SCG2*, *CTSD*, *QPCT*). (C) Gene Ontology enrichment recapitulates these functions: *Program 13* (ER stress, protein folding/ER membrane), *Program 12* (ribosome/translation and ER processes), and *Program 29* (secretory granule lumen, lysosome, peptide inhibitor activity). Dashed lines mark significance thresholds.

We hypothesize that combinatorial codes of broadly expressed endocrine TFs (NEUROG3→NEUROD1, PAX6, NKX2-2) together with tissue-specific TFs (PDX1/GLIS3 for pancreas; CDX2/TBX3 for gut) generate endocrine subtype diversity. A practical test of this model would be to reconstitute these TF combinations in stem-cell-derived endocrine progenitors: (i) *NEUROG3 + PAX6* to establish a generic endocrine state, then (ii) add *PDX1 ± GLIS3* to bias toward a pancreatic $\beta$-like program, or (iii) add *CDX2 ± TBX3* to bias toward an intestinal L/EC-like program. This approach is consistent with prior evidence that *NEUROG3* is necessary for endocrine fate in both pancreas and intestine and that *NEUROD1* reinforces endocrine differentiation [52, 55]. Similarly, *PDX1* and *GLIS3* sustain $\beta$-cell identity and insulin transcription [57, 59, 60], and *CDX2* and *TBX3* participate in intestinal identity and BMP-activated enteroendocrine programs [61, 62].

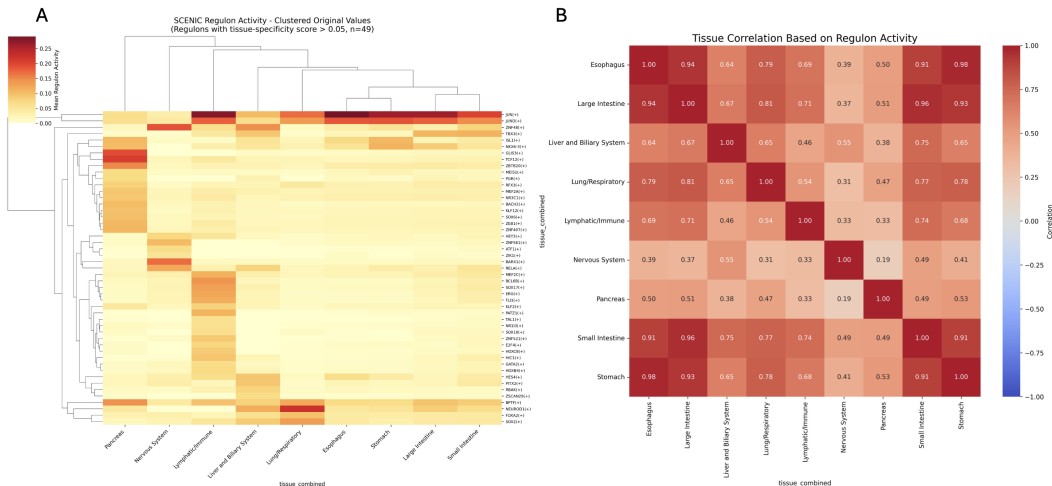

Figure 5: Regulon activity landscape across tissues. (A) Clustered mean SCENIC regulon activity across tissue groups (49 regulons with tissue-specificity score $> 0.05$). (B) Tissue–tissue Pearson correlation matrix computed on the same regulon activity vectors. The arrangement highlights conserved modules (broad, low-to-moderate activity) versus tissue-restricted regulons and recapitulates developmental groupings (foregut stomach–esophagus; proximal–distal intestine) and pancreas-specific control.

# 4 Discussion & Limitations

In this study, we constructed a cross-tissue single-cell atlas of human endocrine cells, revealing key principles of their regulatory architecture. Our primary finding is that endocrine cells are governed by a hierarchical system of both shared and tissue-specific transcriptional programs. A key insight from our cNMF analysis is the identification of conserved pan-endocrine modules related to the high secretory load of these cells, specifically programs governing the ER/UPR and secretory granule biogenesis. This suggests that a fundamental aspect of endocrine identity, beyond hormone production itself, is the maintenance of a robust protein synthesis and trafficking infrastructure. These core programs likely represent a shared functional backbone that is activated and scaled in response to physiological demand across diverse endocrine lineages.

Our analysis further supports a combinatorial logic for endocrine cell identity in which a shared developmental backbone initiated by master regulators like *NEUROG3* is layered with both these conserved functional modules and tissue-restricted transcription factors (e.g., *PDX1* in pancreas, *CDX2* in intestine) to generate adult cellular diversity. These biological insights were enabled by integrating 17 heterogeneous datasets. Our results affirm that deep generative models such as scVI can effectively harmonize data while preserving subtle biological variation critical for rare endocrine populations, thereby providing a robust foundation for downstream network-level analyses (CoVarNet, cNMF) and TF-centric inference (SCENIC).

This atlas aggregates heterogeneous public datasets with uneven tissue coverage and protocol differences; consequently, rare endocrine types are likely under-represented. Stress-response programs, such as the UPR module, may be influenced by tissue dissociation or handling, so attribution to in vivo stimuli remains tentative without time-course data. Integration and decomposition choices (e.g., scVI embeddings, cNMF rank) can alter neighborhood structure and split or merge modules; thus, absolute frequencies and boundaries should be interpreted qualitatively. Network inferences (CoVarNet, SCENIC) are correlative and motif-dependent, nominating regulators rather than proving causality. Finally, this resource is transcriptomic and cross-sectional; spatial, chromatin, proteomic, and perturbation/time-course data will be needed to test the combinatorial TF-code hypothesis.

## 5 Conclusion

In this project, we constructed a comprehensive cross-tissue atlas of human endocrine cells by integrating 17 single-cell RNA-seq datasets. Our analysis successfully navigated complex batch effects to uncover a hierarchical regulatory landscape composed of conserved pan-endocrine programs governing the secretory pathway and tissue-specific modules that reflect distinct developmental origins and specialized adult functions. This atlas serves as a foundational resource for dissecting the systemic coordination of the endocrine system and provides a framework for investigating endocrine dysfunction in health and disease, highlighting fundamental principles of cellular diversification and adaptation across human organs.

## 6 AI agent setup.

We used ChatGPT to brainstorm ideas and plan the manuscript structure, Claude Code to run and iterate on analysis code, and ChatGPT and Gemini to draft and refine text. All model outputs were critically reviewed and verified by the authors; no confidential data were shared with these tools.

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

# A Technical Appendices and Supplementary Material

Technical appendices with additional results, figures, graphs and proofs may be submitted with the paper submission before the full submission deadline, or as a separate PDF in the ZIP file below before the supplementary material deadline. There is no page limit for the technical appendices.

# Agents4Science AI Involvement Checklist

1. **Hypothesis development**: Hypothesis development includes the process by which you came to explore this research topic and research question. This can involve the background research performed by either researchers or by AI. This can also involve whether the idea was proposed by researchers or by AI.

   Answer: blue**[B]**

   Explanation: Human authors defined the core scientific questions (cross-tissue endocrine regulation and regulatory context) and selected the research scope. AI tools assisted with brainstorming alternative framings, surfacing related work, and suggesting candidate hypotheses, but final hypothesis selection, novelty assessment, and scoping decisions were made by the authors after manual literature curation and feasibility checks.

2. **Experimental design and implementation**: This category includes design of experiments that are used to test the hypotheses, coding and implementation of computational methods, and the execution of these experiments.

   Answer: blue**[D]**

   Explanation: Claude generated code for data processing, training, evaluation, and orchestration; ChatGPT executed runs, adjusted configs, and proposed fixes. Human authors provided prompts, objectives, datasets/splits, and acceptance criteria, gave iterative feedback, set seeds and compute budgets, monitored runs, and validated outputs via unit/sanity checks, but did not write code themselves. All artifacts and results were reviewed and approved by the authors, and no confidential or personal data were shared with AI tools.

3. **Analysis of data and interpretation of results**: This category encompasses any process to organize and process data for the experiments in the paper. It also includes interpretations of the results of the study.

   Answer: blue**[D]**

   Explanation: Claude code performed the majority of analysis: ingesting our input data/run artifacts, generating EDA code, aggregating metrics/logs, producing plots/tables, and drafting preliminary interpretations. We supplied ChatGPT with the data/metadata and prompts, and requested literature search/summaries for context.

4. **Writing**: ChatGPT drafted most of the manuscript text (sections, captions, boilerplate) from our prompts and outlines. Gemini was used for cross-checking (proofreading, consistency, citation verification) and style suggestions. Human authors provided the narrative framework and section outlines, reviewed every claim, number, and reference, resolved ambiguities, and finalized figures/tables.

   Answer: blue**[D]**

   Explanation: Large language models drafted $\geq 95\%$ of the manuscript text, captions, and line edits based on our prompts, outlines, and analysis artifacts. Human authors provided the high-level narrative, verified every claim, number, and citation, corrected inaccuracies, ensured consistency with results, and finalized figures/tables. We performed link-level citation checks and unit/metric sanity checks, and we did not provide confidential or personal data to AI tools. The authors accept full responsibility for the final content and compliance with venue policies.

5. **Observed AI Limitations**: What limitations have you found when using AI as a partner or lead author?

   Description: Agentic AI requires explicit, step-by-step guidance; it rarely constructs end-to-end pipelines without users specifying tools (e.g., scVI, Scanpy, CoVarNet), modules, and I/O. Its biological insight tends to be shallow—summarizing patterns rather than proposing mechanistic, novel interpretations. Performance depends heavily on mature, well-documented frameworks; it is weak at inventing new methods or unconventional pipelines. Model quality matters: stronger, instruction-tuned models follow workflows more reliably but still need structured prompts, constraints, and checking. Human expertise remains essential for study design, edge-case handling, statistical validation, and ensuring claims meet publication standards. Finally, we observe account "memory" effects: systems that have accumulated prior context, examples, and iterative feedback behave noticeably better, while fresh accounts without history often underperform until seeded with

scaffolds, datasets, and conventions. Overall, agent AI is a useful accelerator, not an autonomous scientist.

