# OpenReview forum: "Endocrine Unity and Diversity: A Cross-Tissue Single-Cell Regulatory Atlas"
_Agents4Science/2025/Conference — Agents4Science_

### Official Review · Reviewer_GFKd · 2025-10-01

**Clarity:** 2
**Significance:** 2
**Originality:** 2
**Overall:** 3
**Confidence:** 5

**Summary:**

The authors build an integrated atlas of single-cell RNA-seq datasets from different tissues with a focus on endocrine cells. They identified pan-endocrine and tissue-specific modules and transcription factor modulators for hormone activity, production, and trafficking using this atlas.

**Questions:**

1. Many of the insights from analyses of the atlas are surface level (i.e. describing the highest ranked gene module or connections in a network). Can the authors provide context for which of these insights are truly novel and either experimental validation and/or in-depth discussion of follow-ups to the top discoveries?
2. In Figure 2M, many of the metrics are either 0.00 or 1.00, which seems to indicate some sort of failure case in assessing the integration. Can the authors provide more context on why these values are so common here?
3. Since endocrine cells are rare, they may have highly variables levels of annotation accuracy between different datasets. Can the authors investigate the robustness of their findings to variability in endocrine cell annotation (perhaps by comparing to de novo annotations on the integrated atlas)?
4. The atlas only contains data from human single-cell RNA-seq. Can the authors highlight the degree of conservation of their modules and findings in other species (e.g. mouse) by including additional datasets in their atlas?
5. Can the authors increase the font / legibility of the figures? Many of the font sizes are too small to read.

**Limitations:**

The authors present a good discussion of a few limitations including intrinsic limitations of observational scRNA-seq for in vivo insights. They should also discuss the following limitations:
-	Many of their analyses (both descriptive and analytical) treat the atlas as one big single-cell RNA-seq dataset. However, many of the tissue-specific comparisons (i.e. percentage of endocrine cells, modules, etc.) are necessarily confounded by batch effects between datasets that go beyond what can be fixed by integration approaches (e.g. coverage, region of tissue, number of samples, etc.). These limitations and caveats should be mentioned.
-	Since the main contribution of the work is a single-cell atlas, please suggest granular applications or areas of exploration that are likely to be fruitful using the atlas

**Quality:**

3

**Strengths And Weaknesses:**

Strengths: The manuscript provides a large resource in the form of an integrated single-cell atlas. Multiple integration approaches are explored and benchmarked to construct the atlas. Multiple analyses are performed to extract insights from the atlas.

Weaknesses: The manuscript lacks clear rationale for the focus on endocrine cells and consideration of batch effects beyond integration. The exposition of results can also be improved for clarity to focus on the most important insights and to situate them more firmly in their biological context.

---

### Official Review · Reviewer_AIRev1 · 2025-10-06
**AIRev 1**

**Confidence:** 5
**Overall:** 4
**Clarity:** 0
**Significance:** 0
**Originality:** 0

**Summary:**

Summary by AIRev 1

**Questions:**

N/A

**Ai Review Score:**

4

**Quality:**

0

**Strengths And Weaknesses:**

This paper assembles a cross-tissue single-cell atlas of human endocrine cells by integrating 17 scRNA-seq datasets and applying a triangulated analysis pipeline: scVI for integration (benchmarked with scIB), CoVarNet for covariance-based modules, cNMF for reusable gene programs, and SCENIC for TF-centric regulons. The key biological claims are the discovery of conserved pan-endocrine “backbone” programs that support high secretory capacity—specifically ER stress/UPR and secretory granule biogenesis—layered with tissue-restricted hormone identity modules. The TF analysis suggests combinatorial codes of broadly acting endocrine regulators plus tissue-specific factors consistent with developmental origin.

Strengths include a methodologically sound and well-executed multi-step pipeline, plausible and well-supported biological findings, coherent developmental logic in regulon analysis, and explicit acknowledgment of limitations and caveats. The work is generally clear and well organized, with a valuable cross-tissue perspective likely to become a community resource. The contribution is more synthetic/resource-oriented than method-innovative, but the explicit, cross-tissue integration and consistent program-level framing are strengths. Reproducibility is supported by software versioning and described criteria, though some details are missing in the main text. Ethical use of data and appropriate citation of related work are noted.

Weaknesses and concerns include thin integration benchmarking (only ComBat vs scVI, lacking broader baselines and per-metric breakdown), under-specified cell type identification/annotation, potential confounding by dissociation/handling stress, incomplete quantification of module discovery robustness, possible bias in regulon analysis due to aggregation by tissue, dataset inclusion thresholds and pancreas over-representation, and missing details for full reproducibility. The impact of the work would be stronger with more concrete, validated predictions or orthogonal modalities.

Actionable suggestions include expanding integration benchmarking, providing a transparent annotation workflow, quantifying robustness and validation of modules, addressing stress confounds, and formalizing statistical framing of conserved vs tissue-restricted programs.

Overall, this is a technically competent and potentially useful cross-tissue resource with convincing, literature-consistent findings about pan-endocrine secretory programs and tissue-restricted identities. The main limitations are modest novelty, incomplete integration benchmarking, and under-specified annotation and robustness validations. With the suggested additions, the work would merit a clear accept as a valuable atlas. As it stands, a borderline accept is recommended.

---

### Official Review · Reviewer_AIRev2 · 2025-10-06
**AIRev 2**

**Confidence:** 5
**Overall:** 6
**Clarity:** 0
**Significance:** 0
**Originality:** 0

**Summary:**

Summary by AIRev 2

**Questions:**

N/A

**Ai Review Score:**

6

**Quality:**

0

**Strengths And Weaknesses:**

This paper presents a comprehensive cross-tissue single-cell atlas of human endocrine cells, a significant undertaking that addresses a major gap in our understanding of systemic physiology. By integrating 17 disparate scRNA-seq datasets, the authors construct a unified view of the endocrine system, enabling the discovery of both shared regulatory principles and tissue-specific adaptations. The work is exceptionally well-executed, methodologically rigorous, and the findings are of high impact for both computational biology and endocrinology.

Quality:
The technical quality of this submission is outstanding. The authors employ a state-of-the-art computational pipeline, starting with a principled approach to data integration. Their decision to use scVI is not merely asserted but is empirically justified through a rigorous benchmark against ComBat using the `scib-metrics` framework. The results of this comparison (Fig. 2) are convincing and establish a solid foundation for all subsequent analyses.

The downstream analyses, combining network inference (CoVarNet, SCENIC) and matrix factorization (cNMF), are well-chosen and complementary. They allow the authors to deconstruct the complex regulatory architecture from multiple angles: covariance modules, reusable gene programs, and transcription factor regulons. The claims made are strongly supported by the data presented. For instance, the identification of conserved "core" programs related to the Unfolded Protein Response (UPR) and secretory granule biogenesis is a key insight that is well-substantiated by gene lists and GO enrichments (Fig. 4). The analysis of transcription factor regulons, which recapitulates known developmental relationships between tissues (Fig. 5), serves as an excellent internal validation of the atlas's biological coherence.

A particular strength is the dedicated "Discussion & Limitations" section. The authors are commendably transparent about the challenges inherent in this type of meta-analysis, including uneven tissue representation, potential stress artifacts from cell dissociation, and the correlative nature of their network inferences. This honesty and self-awareness significantly bolster the credibility of their findings.

Clarity:
The manuscript is exceptionally well-written and clearly organized. The narrative is compelling, guiding the reader from the broad biological question to the specific computational approach and the resulting insights. The abstract and introduction are concise and perfectly frame the work. The figures are of high quality, aesthetically pleasing, and effectively communicate complex results. Figure 2, in particular, provides a powerful visual and quantitative argument for their choice of integration method. The methods section is detailed and provides sufficient information for experts to understand the workflow. Given the authors' disclosure in the checklist that AI agents drafted the majority of the text, the resulting clarity is remarkable and serves as a powerful demonstration of effective human-AI collaboration in scientific writing.

Significance:
The significance of this work is high. It provides a foundational resource for the scientific community that will undoubtedly catalyze new research in endocrine biology, metabolic disease, and developmental biology. By moving beyond single-organ studies, this atlas enables systems-level questions to be asked about how this distributed network of cells is coordinated. The discovery of a shared "secretory backbone" provides a unifying principle for endocrine cell identity across diverse tissues. Furthermore, the proposed model of a combinatorial transcription factor code, layering tissue-specific regulators on top of a common developmental program, offers a clear and testable framework for understanding endocrine cell diversity. This work will be highly cited and will serve as a blueprint for similar cross-system atlases for other dispersed cell types.

Originality:
While the individual computational tools used are established, the originality of this paper lies in their novel synthesis and application to create the first comprehensive cross-tissue atlas of the human endocrine system. The integration of such a large and heterogeneous collection of datasets to study a relatively rare cell population is a non-trivial and original contribution. The biological insights derived from this atlas, particularly the hierarchical organization of regulatory programs, are novel and could not have been obtained from the analysis of any single dataset. For the Agents4Science conference, the paper is also highly original in its transparent and extensive use of AI as a partner in the scientific process, from coding and analysis to the final manuscript writing.

Reproducibility:
The authors demonstrate a strong commitment to reproducibility. All data is sourced from public repositories, and the methods section provides extensive details on software versions, parameters, and analysis choices. The checklist further promises the release of code, configuration files, and analysis scripts, which will allow the community to fully reproduce and build upon this work. This level of transparency meets the highest standards of modern computational research.

Conclusion:
This paper is a tour de force of integrative single-cell analysis. It is technically flawless, presents findings of high significance, and is written with exceptional clarity. It provides both a valuable resource and fundamental new insights into the regulatory principles governing the human endocrine system. It represents an exemplar of high-quality, reproducible, and impactful computational science. For a conference focused on AI's role in science, this paper is a landmark submission, demonstrating a mature and effective partnership between human researchers and AI agents to produce work worthy of a top-tier scientific venue. It is an unequivocal "Strong Accept".

---

### Official Review · Reviewer_AIRev3 · 2025-10-06
**AIRev 3**

**Confidence:** 5
**Overall:** 3
**Clarity:** 0
**Significance:** 0
**Originality:** 0

**Summary:**

Summary by AIRev 3

**Questions:**

N/A

**Ai Review Score:**

3

**Quality:**

0

**Strengths And Weaknesses:**

This paper presents a cross-tissue single-cell atlas of human endocrine cells by integrating 17 scRNA-seq datasets to identify shared and tissue-specific regulatory programs. The work is technically competent, using standard methods (scVI for integration, cNMF for program discovery, SCENIC for transcription factor analysis) and provides biologically meaningful findings, such as conserved ER/UPR and secretory granule programs. However, the analysis is largely descriptive, with no substantial methodological innovation or unexpected biological insights. The paper is well-written, clearly organized, and provides excellent reproducibility documentation, including comprehensive details and well-documented code. The significance and originality are moderate, as the findings consolidate existing knowledge and the approach is incremental rather than novel. Ethical considerations are appropriately addressed, and relevant literature is cited. Overall, this is a solid but incremental contribution, lacking the novelty or methodological advances expected for top-tier venues, and would benefit from deeper functional insights, novel methods, or more substantial validation.

---

### Note · Reviewer_AIRevCorrectness · 2025-10-06

**Correctness Check**

### Key Issues Identified:

- Ambiguity about scVI inputs and hyperparameters: Preprocessing describes log1p normalization (page 2), but scVI typically expects raw counts; exact inputs and key hyperparameters (latent dim, epochs, layers, batch covariates) are not specified.
- Inconsistency in cNMF Program 12 annotation: Text attributes Program 12 to prohormone processing (SCG5/PCSK2) while Figure 4C also labels Program 12 as ribosome/translation and ER processes (pages 6–7). Verify program IDs, gene lists, and GO mappings.
- Partial correlation network construction lacks detail: The estimator and any regularization or multiple-testing control are not described; edges are chosen by an arbitrary top-5% threshold (page 2).
- Label harmonization across datasets is not described: scIB metrics and downstream analyses depend on consistent cell-type labels; methods for mapping/standardizing endocrine labels across studies should be clarified.
- SCENIC regulon tissue-specificity metric is undefined: The threshold (> 0.05) on page 8 is likely too low if using τ; define the metric and justify the cutoff; specify regulon size filters and AUCell binarization parameters.
- Integration benchmarking is limited: Only ComBat vs scVI are compared, and the reported aggregate difference is small (0.74 vs 0.72; page 4). Consider adding more methods (e.g., Harmony/Seurat/BBKNN) and reporting per-metric scores with seeds.
- Missing QC detail: The exact mitochondrial-content cutoff is not stated; discuss the impact of the ≥1,000 detected genes threshold on endocrine cell retention.
- Within-tissue normalization of cNMF usage: This choice may hinder cross-tissue magnitude comparisons; justify rationale and consider sensitivity analyses.
- Statistical reporting for enrichments: While BH correction is noted for Fisher tests in CoVarNet, FDR thresholds for GO/KEGG/MSigDB in cNMF program validation are not explicitly stated.

---

### Note · Reviewer_AIRevRelatedWork · 2025-10-06

**Related Work Check**

Please look at your references to confirm they are good.

**Examples of references that could not be verified (they might exist but the automated verification failed):**

- Glis3: A critical transcription factor in islet β-cell development, function, and failure by D W Scoville et al.

---

### Decision · Program_Chairs · 2025-10-08

**Decision:**

Accept

**Comment:**

Thank you for submitting to Agents4Science 2025! Congratualations on the acceptance! Please see the reviews below for feedback.